# Effect of the Formation of Amorphous Networks on the Structure and Hydration Characteristics of Granulated Blast Furnace Slag

**DOI:** 10.3390/ma13061462

**Published:** 2020-03-23

**Authors:** Yuhan Yao, Yali Wang, Qi Wei, Suping Cui, Liwei Hao

**Affiliations:** 1College of Materials Science and Engineering, Beijing University of Technology, Beijing 100124, China; b201909030@emails.bjut.edu.cn (Y.Y.); wangyali1978@bjut.edu.cn (Y.W.); qiwei@bjut.edu.cn (Q.W.); 2National Engineering Laboratory of Industrial Big-Data Application Technology, Beijing 100124, China; 3State Key Laboratory of Solid Waste Reuse for Building Materials, Beijing Building Materials Academy of Science Research, Beijing 100041, China; hlw717@126.com

**Keywords:** amorphous structure, granulated blast furnace slag, cement, MAS NMR, hydration activity

## Abstract

The slag obtained in the process of pig iron smelting has been widely used, but the variational hydration activity always is a significant factor affecting its quality. In this experiment, the laboratory simulated slag was prepared by adjusting the chemical composition and cooling method. The experiment primary characterized the structure and hydration process with different types of slag by using MAS NMR, XRD, compressive strength, ICP, SEM, and hydration heat, then obtained the influence of the composition of the network former S/A (the mass ratio of SiO_2_ and Al_2_O_3_ in chemical composition) and amorphous phase content on its structure and hydration activity. The result shows that lowering the S/A value can reduce the degree of vitreous polymerization in the slag; reducing the S/A value of the slag can make the slag hydration time advance, and consequently, the cumulative exotherm increases, the liquid phase Ca/Si and Al/Si ionic ratio increases, and the hydration product changes from C–S–H gel to C–A–S–H gel, which ultimately leads to an increase in compressive strength. In the high S/A value slag, the formation of the trace crystal phase of gehlenite is beneficial to reduce the degree of polymerization of the amorphous.

## 1. Introduction

The granulated blast furnace slag (abbreviated as slag or GGBFS) produced as a by-product in the process of pig iron smelting is a low-cost, environmentally-friendly, and energy-saving material with high working performance. Due to its potential hydration activity in the alkaline environment, the performance of this kind of solid waste as a building material can be comparable to Portland cement and has superior durability [1,2]. According to a specific investigation of the steel import and export report, China’s pig iron output in 2017 was 711 million tons with sustained and steady growth [3]. Based on the estimation that 0.3–1 tons of slag yield per ton of pig iron, the potential GGBFS treatment capacity of the whole Chinese industrial market is about 200 million tons still [4,5].

Plenty of factory and laboratory data show that the hydration activity of GGBFS always displays an oscillatory property in broad scope whether the chemical composition is similar or not and this problem, to some extent, limits the quality level of product, which seriously affects its application and brings the subsequent inevitable process involving economic, energy, and labor consumption [6,7,8]. The utilization of slag in cement-based materials mainly depends on its hydration activity and it cannot be used reasonably and effectively if its activity is not accord with the standard. Therefore, improving product quality and application value is still the core issue in the treatment of blast furnace slag [9].

Evaluating the hydration activity and quality of GGBFS from the perspective of chemical composition has been the primary evaluation criterion by the empirical exploration in the industrial production process and laboratory investigation over past years. The main chemical composition of slag is exhibited in the form of oxides (wt.%): 30%–45% CaO, 25%–40% SiO_2_, 5%–25% Al_2_O_3_, 5%–10% MgO, and trace impurity oxides dominated by TiO_2_ and MnO [10,11,12]. A series of typical chemical-composition-based hydration activity coefficients (such as the quaternary basicity coefficient, Si/Al ratio, etc.) can roughly predict the hydration activity of some GGBFS, but a display simultaneously inaccurate prediction for others, and sometimes even obtains a significant deviation [13,14,15,16]. The leading causes for this phenomenon are that, firstly, GGBFS, with the same chemical composition and fineness, behaves differently in mineral composition (the content of the amorphous phase and types of crystalline phase) as a result of variable cooling methods [17,18]; secondly, the difference in polymerization degree of network former in amorphous phase structures also significantly affects the hydration activity of GBFS [19]. Therefore, combining the aspects of chemical composition, mineral composition, and amorphous structure to carry out a complementary macroscopic and microscopic analysis of the structure and hydration characteristics of blast furnace slag is viewed as an effective total solution to the problems proposed previously mentioned [20].

In the study of the structure of blast furnace slag, a large amount of research has shown that the formation of an amorphous glass phase network is mainly the polymerization process of network formers [SiO_4_]^4−^ and [Al_2_O_4_]^5−^ through the connections among them by the bridged oxygen under quench conditions [19,21,22,23,24]. The structural characteristics of the slag can be characterized by the bridged oxygen number, which can be obtained quantificationally and qualitatively by ^29^Si and ^27^Al MAS NMR [25,26]. As with nearly two decades of the rapid development of solid nuclear magnetic technology, especially for MAS NMR, which has been applied to the field of silicate building materials, researchers have conducted a large number of preliminary studies on the structure of vitreous in slag [27,28]. Significant advances include the coordination states of elements in a series of geopolymers (dominated by slag, fly ash, metakaolin, and so forth) and the possible bridging patterns of these ligands [29,30]; the role of ligands and metal cations in the formation of amorphous phase networks [31,32]; the structural variational characteristics of slag during hydration; and of gel-like hydration products [33,34]. Researches generally take industrial slag as the samples by which several inevitable effects come out. First of all, it causes chemical composition and structure variability and uncontrollability. In addition, the regularity of quantitative analysis reduces in the process of analyzing the structural formation of amorphous phase networks. Finally, it is difficult to analyze the formation characteristics of amorphous phase networks based on chemical composition precisely because of the complex and numerous inducements for network modification. On the whole, structure detection technologies and means for slag are quite complete, but such technologies are not systematically used to detect the slags that have regularly varying chemical compositions. The hydration activity of slag is not only related to the chemical composition or to the structure, but it should be analyzed as a correlative system.

In this study, the production process of industrial slag was simulated with a controllable composition of different S/A values by using pure chemical reagent as a raw material undergoing the air-quenched process [35]. Accurate characterization of the mineral composition and amorphous structure of slag by ^29^Si and ^27^Al MAS NMR and XRD—combined with hydration characteristics—complements the lack of merely predicting the activity of slag by chemical composition. The ultimate purpose of this research is to provide a theoretical basis and technical support for the more precise quality prediction and control of slag.

## 2. Materials and Methods

### 2.1. Preparation Process of Air-Quenched Slag

#### 2.1.1. Raw Materials

In order to keep the controllability of the formation conditions of slag and the factors affecting the formation of the amorphous phase, the preparation of slag was strictly designed with high purity chemical reagents, including chemically pure SiO_2_, Al_2_O_3_, CaO, MgO, and BaO, produced by Beijing Chemical Works Co., Ltd. (China), with a purity of ≥98.0%. The addition of a trace of BaO as the fluxing agent can adjust the viscosity of molten slag in order to control the amount of non-crystalline phase in the mineral composition. Meanwhile, the effect of BaO on the formation of the amorphous phase is also discussed in the following sections. All the chemical reagents used for the preparation of slag were weighed by an electronic balance (ME204, Metler-Toledo, Columbus, OH, USA) to ensure the accuracy was within 0.1 g, and the mixed reagents were sufficiently stirred in the mixer over 1 hour to approach uniformity. 

#### 2.1.2. Chemical Composition Design

The structural characteristics of the network in the amorphous phase are mainly affected by the polymerization of network formers, of which, especially, the types and proportions. The main network formers in the slag were aluminum–oxygen tetrahedrons [Al_2_O_4_]^5−^ and silicon–oxygen tetrahedrons [SiO4]^4−^ and the quality ratios between different network forming bodies were noted as S/A. Table 1 shows that the reasonable design guideline of slag samples in this research is the ratio between network formers and network modifiers (C + M/S + A); the ratios between different network modifiers (C/M) in the chemical composition were rigorously fixed, along with S/A as the variable. Meanwhile, C, M, S, and A represent the mass percentage of CaO, MgO, SiO_2_, and Al_2_O_3_.

Based on the chemical composition of SA3 and SA8 in Table 1, a small amount of BaO was added to reduce the viscosity of melting slag and control the amorphous content, and the chemical composition of designed samples is shown in Table 2.

#### 2.1.3. Quenching Granulation and Powder Grinding Process

Referencing the production process adopted from industrialized technology, the uniform raw materials were heated from room temperature to a maximum, ultimate temperature of 1500 °C under the conditions of the air environment, followed by heat preservation for 30 min, before being dumped into the fan blade granulating device as shown in Figure 1 to take the quenching granulation [36]. The final obtained slag particles were ground on a horizontal planetary ball mill to a specific surface area of 450 m^2^/kg in order for the physical properties to correspond with GB/T 203-2008 [37].

### 2.2. Methods

#### 2.2.1. Mineral Composition Analysis

After passing the finely ground slag powder through a sieve with a hole size of 0.08 mm, the powder under the sieve was taken for X-ray diffraction (XRD) analysis (D8 ADVANCE X-Ray Diffractometer, Bruker, Karlsruhe, German) to obtain the phase composition. The operating parameters were in continuous scanning mode, the range of diffraction angle was 5°–70°, Cu target, with a scanning step size of 0.02°. XRD analysis of hydration products required pretreatment of the samples, which included placing the cracked sample in anhydrous ethanol to guarantee the internal hydration process was terminated, followed by desiccation in a vacuum drying oven under 40 °C over 48 h. To conduct a subsequent XRD analysis, it was necessary to grind the fragments in an agate mortar and separate them according to the same sieving conditions as the slag powder described above.

According to GB/T 18046, the ratio of the area of the smooth bread peak (representing the amorphous phase) and the area of the sharp crystalline phase diffraction peak in the range of the diffraction angle from 22° to 38° in the XRD pattern was taken as the ratio of the content of the amorphous phase to the content of the crystalline phase [38].

#### 2.2.2. Hydration Activity Analysis

Paste with 50% of each slag and 50% OPC (PI 42.5) of W/C = 0.35 was shaped in a 20 mm × 20 mm × 20 mm mold which referenced the method of GB/T 17671 [39]. The sample was firstly cured in the curing chamber with a constant temperature (20 ± 1 °C) and humidity (90% ± 5%) stated for one day, then demolded and water-cured on day 3, day 7, and day 28, testing the compressive strength of the obtained samples, respectively.

#### 2.2.3. Early Ion Dissolution Characteristics Analysis

Five grams of slag powder were blended with 20 g 0.1 mol/L NaOH solution by stirring with a magnetic stirrer to reach the reaction time of 5 min, 30 min, 1 h, 4 h, 8 h, 12 h, and 1 d. Two grams of the filtered supernatant mixture were taken with 18 g water and 2 g 0.1 mol/L nitric acid for dilution (11 times) and acidification treatment (pH ≤ 7). Then, the concentration of Ca, Si, Al, and Mg ions in the filtrate was quantitatively analyzed by ICP (Optima 5300DV, Perkin Elmer, Fremont, CA, USA).

#### 2.2.4. Microstructure of Hydration Products Analysis

Epoxy resin was used to cover the sample fragment obtained from the treatment in Section 2.2.1 that had been nonreactive and dry, followed by grinding and polishing using sandpaper with specifications of 400, 800, 1200, 1600, and 2000 in sequence. The polished samples with smooth sections were used during backscattering SEM (S-3400N, Hitachi, Tokyo, Japan) to observe the phase composition, phase distribution, and the hydration ring’s thickness of hydrated slag, and the unpolished samples with appropriate sizes were used in the observation under a secondary electron scanning mode in which the microscopic surface morphology was observed.

#### 2.2.5. Structure Analysis of the Amorphous Phase

The chemical shift of ^29^Si in MAS NMR (Avance III 400M, Bruker, Karlsruhe, German) spectra depends on its nuclear environment, which mainly depends on the degree of polymerization between the network formers. According to the classification of the bridge oxygen number (n) formed by several silicon–oxygen tetrahedrons, the polymerization state (Q^n^) and corresponding chemical shift of the silicon resonance were listed, respectively: Q^0^ at −68 to −76 (ppm), Q^1^ at −76 to −82 (ppm), Q^2^ at −82 to −88 (ppm), Q^3^ at −88 to −98 (ppm), and Q^4^ at −98 to −129 (ppm) [40,41]. If the silicon–oxygen tetrahedron linked to an aluminum–oxygen tetrahedron, the chemical shift of the formed Q^n^_(mAl)_ increased by about 5 ppm relative to Q^n^ [42,43]. ^27^Al MAS NMR mainly distinguished the coordination number of Al ions in the slag, among which the chemical shift of 4,5,6-coordinated aluminum resonances were 80–50 ppm, 40–30 ppm, and 15–10 ppm, respectively [44,45]. The pulse program used high power decoupling (hpdec) and the operating parameter of MAS NMR in this experiment is shown in Table 3.

Research has shown that the intensity of the characteristic peak is directly proportional to the number of ^29^Si nuclei in different environments, so that the concentration of silicon–oxygen tetrahedrons in different polymerization states can be obtained through the deconvolution of the spectra [46]. The Lorentz–Gauss method is used for the deconvolution in this paper.

#### 2.2.6. Exothermic Characteristic of Hydration Process

The monitoring of the exothermic characteristic of the paste during hydration based on ASTM C1679-17 is conducted using a thermal analysis meter (TAM AIR, TA Instruments, New Castle, DE, USA) [47]. The mix proportions and water–cement ratio are the same as those in the compressive strength test. After mixing the paste in the ampoule with an internal stirrer, the heat release in the hydration process produced in the following 3 days was monitored in real-time with a step width of 20 s.

## 3. Results

### 3.1. Analysis of Slag with Different S/A

#### 3.1.1. Mineral Composition

Figure 2 shows the mineral composition of slag with different S/A, and quantitative analysis. The distinct steamed-bun-shaped peaks in the XRD patterns of SA1, 3, and 6 indicate that the mineral composition basically presents as an amorphous phase, while the amorphous content in SA7 is significantly reduced with the increasing intensity of diffraction peaks of crystalline phases with low activity such as gehlenite (Ca_2_(Al(AlSi)O_7_)), akermanite (Ca_2_Mg(Si_2_O_7_)), manganolite (Ca_3_Mg(SiO_4_)_2_), and melilite (Ca_2_(Mg_0.5_A_l0.5_) (Si_1.5_A_l0.5_O_7_)).

#### 3.1.2. Amorphous Phase Structure

Figure 3a shows the ^29^Si MAS NMR spectra and deconvolution results of the slag with different S/A. The signal at −68, −74, and −82 ppm can be attributed to Q^0^, Q^1^, and Q^2^ silicon atoms, respectively. Under the condition of fixed C/M and (C+M/S+A), the amount of Q^2^ decreases with a decreasing S/A in chemical composition, as illustrated by the comparison between the spectra of SA3 and SA6. The Q^2^ signal can be separated into Q^2^_(Al)_ and Q^2^ with no oxygen bridge link to [AlO_4_]^5−^ tetrahedra. Q^1^_(Al)_ cannot be detected in all samples.

Al^3+^ is generally believed to act as a network former by four-coordination (Al IV) and network modifier by six-coordination (Al VI) in slag. Researchers have pointed out that the chemical shift of Al(IV) is at (50 ± 20 ppm) and that of Al(IV) is at (0 ± 10 ppm). According to Loewenstein’s exclusion principle, [AlO_4_]^5−^ tetrahedra do not link directly to other [AlO_4_]^5−^ tetrahedra, and a bridging [SiO_4_]^4−^ tetrahedron is required between them. Therefore, in slag, asymmetric Si–O–Al linkages are more favorable than symmetric Al–O-Al structures. Each aluminum–oxygen tetrahedron was connected strictly with n silicon–oxygen tetrahedrons through bridged oxygen, while reserving (4−n) non-bridging oxygen.

As can be seen in Figure 3b, the existence of a broad peak at 60 ppm implies that the slag aluminum has a coordination number of four located in the highly distorted tetrahedral (Al IV), regardless of the change in S/A. The result shows that Al^3+^ behaves rather as a network former with the silicon–oxygen tetrahedron than as a network modifier located among the network. The partial replacement of Si^4+^ by Al^3+^ in the tetrahedron leads to charge imbalance, and therefore, metal cations are required for charge compensation at the central position and the endpoints of [AlO_4_]^5−^ tetrahedra. The influences of changing S/A can be summarized as follows:Influence the mineral composition;Change the polymerization degree of network formers;Rearrange the distribution of metal cations located in the network.

Consider Figure 4, which plots a schematic description of the difference between slag with higher S/A and with lower S/A. Initially, cations tend to be enriched around the aluminum–oxygen tetrahedrons due to their charge imbalance because of the high negative charge characteristic of the aluminum–oxygen tetrahedron that is directly connected to the silicon–oxygen tetrahedron. Subsequently, the Q^n^ in the structure changes from Q^1^ and Q^2^ to Q^0^ by the increasing modification effect of the attracted metal cations, and the enrichment of metal cation in the area of the dashed line was weakened for the more widely distributed negative form charge in the structure. The polarization effect of O in the [AlO_4_]^5−^ tetrahedron is stronger than in the [SiO_4_]^4−^ tetrahedron, which lengthens the bond length of Al–O than Si–O, accompanied by bond energy decreasing, ionic bond characteristics increasing, and covalent bond characteristics decreasing, making it easier to hydrolyze in an alkaline environment. On the contrary, with the decrease of S/A, the content of the crystalline phase in mineral composition increases while the active amorphous phase decreases. Thus, there exists a trade-off between the structure and the mineral composition for higher hydration activity.

Up to now, much research on phase separation of glass in slag has been done, and two kinds of phase separation of silicon-rich and calcium-rich structures can be observed by a transmission electron microscope (TEM). The higher the content of the calcium-rich phase in the amorphous phase, the higher the activity of slag and hydration rate in an alkaline environment. As mentioned above, the proportion of net former is closely related to the forming mechanism of the amorphous one, which is in good agreement with the generation mechanism of phase separation structures.

#### 3.1.3. Compressive Strength Development of Paste

Figure 5 presents the compressive strength development of slag-cement paste with different S/A at 3 d, 7 d, and 28 d. SA1 gives a strength of 15 MPa, 20 MPa, and 37 MPa, respectively, which remains relatively stable as S/A ranges from 5 to 3.5. The same regularity also appears in SA4, SA5, and SA6, of which the 3 d, 7 d, and 28 d compressive strength increases to 18 MPa, 25 MPa, and 42 MPa compared with SA1. For slag with S/A below two, the compressive strength markedly increases to 23 MPa, 35 MPa, and 46 MPa, which indicates a significant improvement of hydration activity.

As mentioned above, the compressive strength of paste increases with the decrease of S/A when other structural factors are constant. There is a contradiction between the mineral composition and the hydration activity that SA1 with the highest amorphous content showed the lowest activity. One possible solution to this problem is that the structural characteristic is the main factor to determine the potential hydration activity.

Combined with the relation between the structure of amorphous and paste compressive strength, the analysis shows below:Higher S/A slag contains a high concentration of Q^1^ and Q^2^ species, which need to consume a large amount of OH^-^ during the hydration process to destroy the structure and lead to reduced activity effectively.Although the lower S/A slag forms a more crystalline phase, the formed amorphous phase has a relatively lower degree of polymerization and is more activated to hydrolysis, so it exhibits higher compressive strength. However, such slag can provide less active substance, so its activity may fall behind the slag of higher S/A after prolonged hydration age.

#### 3.1.4. Hydration Kinetics

Figure 6 depicts the normalized heat flow and heat release of different S/A slag during hydration. The hydration process of slag can be divided into three stages: the early dissolution stage (<30 min), the transition stage (30 min to 5 h), and the hydration stage (>5 h) [48]. The curve of SA1 has no visible slag hydration exothermic peak (Figure 6a), indicating that the hydrolysis of the slag is slow during hydration. This situation is not conducive to the early strength development of slag, but from the perspective of cumulative exotherm, the exotherm of the paste at 60 h can reach 175 J/g, which is at a higher level in each group of samples. As S/A decreases, a generally recognized hydration peak of slag appears in the normalized heat flow curve, and the initial hydration time gradually advances, while the maximum instantaneous heat flow rises. The most active SA1 reaches the hydrolysis condition after five hours and begins to release heat intensely in which the maximum heat flow could reach 3.25 mw/g, but the reduced cumulative heat of 138 J/g is significantly lower than the slag with higher S/A. 

The slag hydration is the destruction of the amorphous phase network by OH^−^ [49]. For a higher SA value slag (SA > 3), the amorphous structure has a high degree of polymerization, so it is difficult to hydrolyze in an alkaline environment rapidly and release noticeable heat instantaneously.

When SA < 3, the degree of polymerization of the amorphous phase structure decreases, in which there is an increasing amount of Si–O bonds and Al–O bonds with lower energy. As a result, the normalized heat flow increases, and the starting hydration time moves up. However, the content of the amorphous available for hydration is reduced, which causes a decrease in the cumulative heat release.

The slag with an S/A equal to 2.4 is both well-behaved on heat flow and accumulated heat release, indicating SA5 has considerable content of amorphous phase with a low level of polymerization, which is beneficial to early and long-term age hydration.

#### 3.1.5. Early Ion Dissolution Characteristics

For a visual representation of the variation of the concentration of main ions Ca^2+^, [SiO]^n−^, Al^3+^, and Mg^2+^ released by slag in the liquid phase with time (<3 day) during hydration, the reader is referred to Figure 7.

As shown in Figure 7a, the dissolution characteristics of Ca^2+^ in an alkaline environment mainly has two stages. Initially, the Ca^2+^ on the surface of slag particles which is not encapsulated by the amorphous network is released to the liquid phase directly and shows a rapid concentration increase in the early stage. Afterward, [Ca^2+^] decreases as hydration proceeds and the formation of hydration products. With the continuous hydrolysis of slag, the [Ca^2+^] slowly rises and stabilizes after one day. After hydration in an alkaline environment for 72 h, the [Ca^2+]^ in the liquid phase of the low SA value sample (SA5,7,9) was significantly higher than that of the high SA value sample (SA1/3), of which SA1 with highest S/A shows a lowest [Ca^2+^] only for 15 mg/L.

From Figure 7b, it can be seen that the [SiO]^n−^ creeps up first and then decreases due to the relative high Si–O bond strength in the amorphous network. In conclusion, the higher the S/A value of slag, the higher the [SiO]^n−^ in an alkaline environment.

The variation of [Al^3+]^ in an alkaline environment changes with time is illustrated in Figure 7c, which shows a sizable dissolving-out amount prevailing over other ions during the whole hydration process. Slag with lower S/A (SA5, 7 and 9) contains more Al^3+^, and provides more resources than that of the high S/A sample (SA1 and 3), so the curve for SA9 is much higher than SA1. Coincidentally, the trend of [Ca^2+^] and [Al^3+^] has obvious synchronism, which is caused by the discussion in Section 3.1.2 that the location of the aluminum–oxygen tetrahedron in the amorphous phase is easily enriched by more metal cations.

However, there is no measurable change of [Mg^2+^] during the whole hydration process.

#### 3.1.6. Structural Evolution of Slag During Hydration

Figure 8 suggests the ^29^Si MAS NMR spectra and the deconvolution of SA1, 3, 6, and9 after hydration for three days in an alkaline environment. Compared with the MAS NMR spectrum of the unhydrated slag in Figure 3, it can be observed that a series of new peaks are detected at a chemical shift of −72, −75, −80, −85, −89, and −90 ppm, which represent Q^1^, Q^1^_(product)_, Q^2^_(Al)_, and Q^2^, respectively. 

As reported by other researchers, Q^1^_(product)_, Q^2^(Al), and Q^2^ mainly represent the silicon–oxygen tetrahedron on the hydration product of C–S–H gel [50]. Table 4 summarizes the quantitative analysis of Q^n^ in the paste of variable slag, and shows that, for SA1, the hydration product contains 32% of Q^2^ and 41% of Q^2^(Al), which are significantly lower than the values of 12% and 68% of SA9, respectively. This comparison indicates that the hydration product in the paste is converted from the C–S–H gel to the C–A–S–H gel as the S/A value decreases.

Additionally, the degree of hydration (DH) of slag can be calculated according to the decrease of the amount of (Q^0^ + Q^1^) after the hydration. The DH of SA1 is 54% at 28 days’ age, whereas the DH of SA9 is 65%. It is evident that the network of amorphous can be more easily destroyed in the alkaline environment with a decrease of S/A. 

#### 3.1.7. Hydration Products and Micro-Morphology

As follows from the Figure 9 shown below, the hydration products at 28 days’ age are mainly Ca(OH)_2_, and gels. For SA3 (S/A = 3.5), C–S–H gel is formed and would be replaced by C–A–S–H gel when S/A changes to 1.8.

The illustration of micro-morphology is also identical with the XRD analysis. For SA3 (S/A = 3.5), there is only a 2 μm wide thin hydration ring around the slag particle, but for the SA6 having an S/A value of 1.8, a distinct hydration ring can be observed. The EDS results of the hydration ring show that the atomic percentage of the material in this region is: 68% of O, 6% of Mg, 4.6% of Al, 8.2% of Si, and 13.3% of Ca, which can be considered as a zeolite-based gel.

The formation of the plate-like hydration zone around slag particles can be observed in Figure 10a,d, which is indeed a dense region that has excellent mechanical performance and is formed by zeolite-based gel, also known as C–A–S–H gel, coating crystalline minerals [51]. As shown in Figure 10e,f, a large amount of columnar scolecite can be observed in the hole left by the slag particles falling off, and a good deal of cuboid thomsonite and flaky heulandite cover the surface of the slag. The slag hydration ring is a dense region formed by zeolite-type amorphous C–A–S–H gel-coated crystalline minerals with high mechanical properties [52]. 

The micro-morphology of SA3 with S/A = 1.8 is shown in Figure 10b,c. Compared with SA6, the plate-like hydration zone is substituted by an incompact gel-like substance. By the same token, the amount of zeolite-based minerals in the remaining hole significantly decreases. Given these points, the analysis results of the SEM image also explain the regularity of the compressive strength of the paste decreasing with the S/A value of the slag.

### 3.2. Analysis of Slag with Different Contents

#### 3.2.1. Mineral Composition

The XRD patterns of the slag with varying amorphous content are presented in Figure 11. As shown in Figure 11a, since G1 and G5 both have the highest melting point and the highest viscosity in a series of slags of uniform chemical composition, the formation of many inactive minerals such as gehlenite (Ca_2_(Mg_0.5_Al_0.5_) (Si_1.5_Al_0.5_O_7_)), melilite (Ca_2_(Mg_0.5_Al_0.5_) (Si_1.5_Al_0.5_O_7_)), and merwinite (Ca_3_Mg(SiO_4_)_2_) can be observed. 

As quantitative calculations for the slags of S/A = 1.8, C/M = 5.5, and (C+M)/(S+A) = 1.03, the amorphous content of each sample and the mineral content of each crystal phase change with the decrease of melting point as shown in Figure 11b. As the melting point of slag decreases, the amorphous content in slag increases. Notably, the amorphous content of G4 can reach more than 95%. When S/A = 3.5, C/M = 5.5, and (C+M)/(S+A) = 1.03, the amorphous content of the G5 sample with the highest melting point has already reached more than 95%. Owing to this, lowering the melting point cannot cause significant changes in the amorphous content of the slag in this case.

#### 3.2.2. Amorphous Phase Structure

Figure 12 depicts the change in its structure as the amorphous phase content changes. According to the ^29^Si MAS NMR spectra of G5 and G8, the chemical shift of the silicon–oxygen tetrahedron in G5 is mainly concentrated between −60 and −80 ppm, corresponding to the state of Q^0^, Q^1^, and Q^2^, with concentrations of 15%, 50%, and 34%, respectively, when the glass content is 90% (Table 5). For the G1 and G4, the decrease in melting point only causes an increase in the amorphous phase content but is irrelevant to the network structure.

Comparing Figure 12a,b, the silicon oxide tetrahedrons in the form of Q^0^ and Q^1^ are converted to the Q^1^ and Q^2^ forms when the amorphous phase is increased, indicating that the degree of polymerization of the silicon tetrahedron in the amorphous phase is increased, and the amount of bridge oxygen increases. As a result, more OH^−^ are needed to destroy the network during the hydration process to make metal cations effective release. Q^0^ disappears substantially in G8, and the content of Q^1^ and Q^2^ reaches 50% and 49%, respectively, when the amorphous phase reaches 98% or more (Table 5). 

The incorporation of BaO in the experiment only acts as a flux to reduce the melting point of the slag. In order to discuss the relationship between the amorphous content, structure, and hydration activity, it is necessary to eliminate the influences of the incorporation of BaO. First of all, the amount of BaO in the experiment is minimal, and the maximum amount is only 0.75% of the total mass. Furthermore, from the comparison of Figure 12a, it does not increase the degree of polymerization of the amorphous phase structure. Moreover, the trend of reducing the degree of polymerization of structure also can be ignored, see Figure 12b. Consequently, the reason for the change of the amorphous phase structure in G8 is the increase of its content rather than the incorporation of BaO, while the incorporation of BaO only affects the mineral composition of the slag.

#### 3.2.3. Compressive Strength Development of Paste

The compressive strength development of paste mixing G1–G4 is as Figure 13 represents. The value reaches 22 MPa, 34 MPa, and 46 MPa at 3 d, 7 d, and 28 d when the amorphous content is 73%, and increases by 9 MPa, 18 MPa, and 32 MPa, respectively, with an increment of 12% amorphous content. The same growth trend is also reflected in G5−G8, but the increase in 5% amorphous phase content only causes the compressive strength to a comparatively low increase at different ages, about 4 MPa, 14 MPa, and 20 MPa.

As the content of the amorphous phase increases, the hydration activity of the slag shows an increasing trend, but the growth rate is related to the chemical composition of the slag itself. The hydration activity of the slag rarely increases with the increase of amorphous content if it has reached 95% or more. Conversely, the hydration activity of G7 and G8 shows a specific trend that as the amorphous content continues to increase, the hydration activity decreases slightly when the amorphous content has reached 95%. The reason can be explained by the fact that the increase of amorphous content increases the degree of polymerization on the network structures. 

Given these points, the slag is easy to form a high degree of polymerization amorphous phase and, the presence of trace crystal phase is beneficial to the improvement of slag activity; while for a low S/A value of slag, it is not easy to form a high degree of polymerization of tetrahedrons for its structural characteristics, and for the effective use of such slag, it should be ensured that the amorphous content can reach the maximum allowable by the process.

## 4. Discussion

According to the results above, the regular change of S/A results in a variation of the degree of polymerization of the structure of the amorphous network, which shows the influence on hydration activity development, heat release, ion dissolution characteristics, and the type of hydration product.

In this paper, the qualitative and quantitative analysis of [SiO_4_]^−^ and [AlO_4_]^-^ by the method of MAS NMR spectra are in good consistency with previous research [25,26,27,28,29,30]. The effects of S/A, C/M, and (C+M)/(S+A) to the structural and hydration characteristics of slag are often studied together [11,12,13], but in Section 3.1 and Section 3.2, the experiment focused only on S/A and amorphous phase content, by both of which trying to study the effects of the formation of amorphous networks. The content of SiO_2_ and Al_2_O_3_ have been studied as a chemical composition problem in much research, and the slags with high S/A value usually result in a low degree of polymerization and a low hydration activity [11,12,22]. In this paper, the results are consistent with the previous at an early age of hydration as Section 3.1.3, Section 3.1.4, Section 3.1.5, Section 3.1.6 and Section 3.1.7 shows. However, at a longer age, the difference in compression strength of different slags is narrowed down but still exists, which is caused by the transformation of the type of hydration products. In summary, the S/A value mainly affects the compression strength and heat release rate of slag-cement at an early age rather than during the whole period. Although slag with high S/A values has excellent strength development, its intense heat release at an early age increases the risk of cracking, which is a noteworthy problem in engineering.

The content of the amorphous phase affected by the viscosity and cooling rate is the main influencing factor affecting the long-age hydration activity and is universally considered to be proportional to it [10]. However, for the slag having a high S/A value (>3.5), a small quantity of crystalline phase reduces the degree of polymerization, and has little effect or even has a positive effect on the hydration activity as Figure 13 shows. In the actual industrial process, it should be considered that the viscosity and cooling rate could be controlled to reach a specific value rather than the maximum.

## 5. Conclusion

This paper mainly researches the influence of changing S/A and amorphous content on the amorphous phase structure and mineral composition, also analyzes the hydration characteristics and mechanism of different kinds of slag. The results show as below:The decrease of S/A value in chemical composition can effectively reduce the degree of polymerization in slag, as evidenced by the change of Q^2^ to Q^0^ and Q^1^, which is accompanied by the homogenization of metal cations distribution among the network. However, it increases the content of the crystalline phase in the mineral composition.The degree of polymerization of the amorphous phase network in the slag decreases with the S/A value. This phenomenon causes the advance of slag’s hydration starting, and the cumulative heat release increases, the Ca/Si ratio and the Al/Si ratio in the liquid phase increase, the hydration product converts from C−S−H gel to C−A−S−H gel, and finally causes the increase in compressive strength through the macroscopic perspective.The regularity that the compressive strength rises with the increase of S/A is more evident in the early stage of hydration. Increasing the S/A value of 3.5 can increase the three-day and seven-day strengths of the slag-cement paste by 67% and 85%, respectively. However, for the longer age, the content of the active amorphous phase is reduced due to the high crystal phase content in high S/A value slag, so the 28-day hydration activity increase is only 36%, and the differences in the activity is gradually reduced.Slag with high S/A tends to form a highly polymerized amorphous network structure. When the content of the amorphous phase is higher than 95%, the degree of polymerization of amorphous increases as the content of it sequentially increases, resulting in a small decrease in the hydration activity. In contrast, the low S/A value slag rarely forms an amorphous phase with a high degree of polymerization, and the compressive strength of paste is completely proportional to the content of amorphous.

## Figures and Tables

**Figure 1 materials-13-01462-f001:**
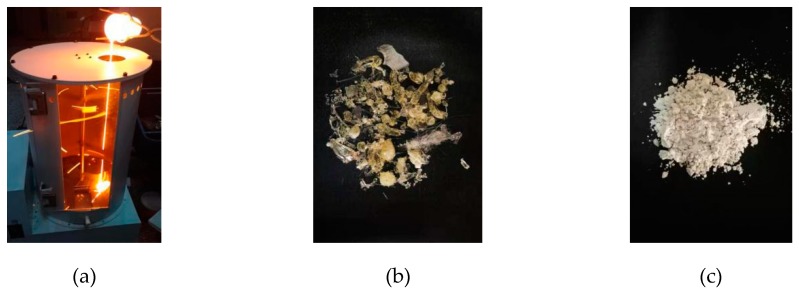
Design of the fan blade granulating device: (**a**) the realistic snap of quenching granulation; (**b**) the final slag product; (**c**) slag powder with specific surface area of 450 m^2^/kg.

**Figure 2 materials-13-01462-f002:**
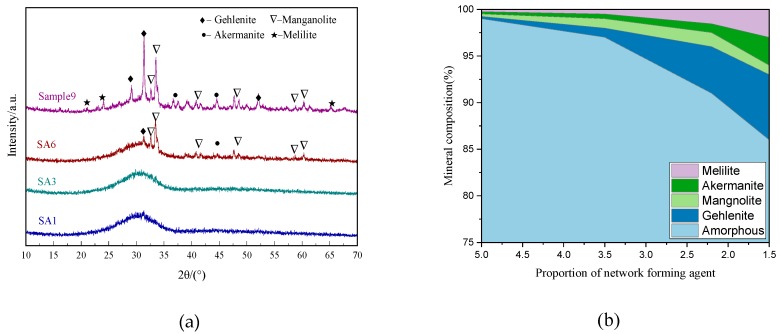
Mineral composition of blast furnace slag with different S/A: (**a**) XRD patterns; (**b**) quantitative analysis.

**Figure 3 materials-13-01462-f003:**
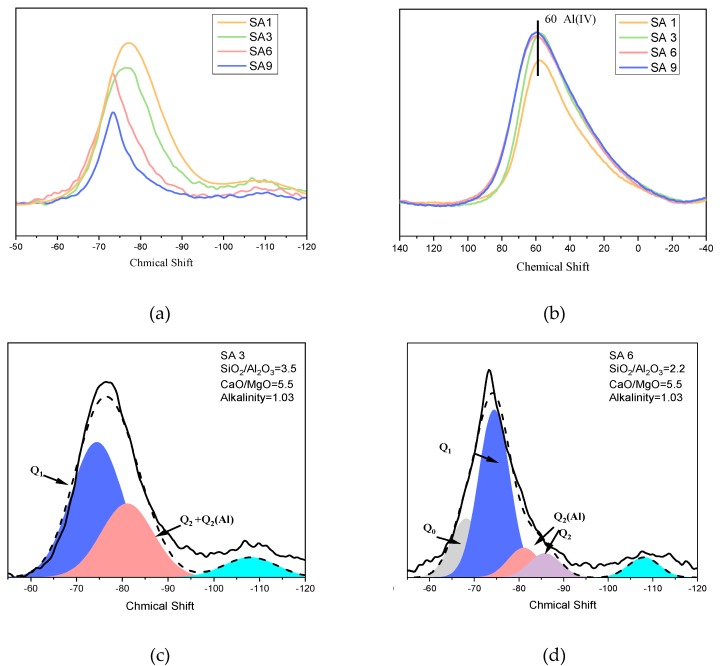
^29^Si MAS NMR spectra of slag with different S/A ratio: (**a**) unhydrated SA1, SA3, SA6, and SA9; (**b**) ^27^Al MAS NMR spectra of slag with different S/A; (**c**) the deconvolution of unhydrated SA3; (**d**) the deconvolution of unhydrated SA6.

**Figure 4 materials-13-01462-f004:**
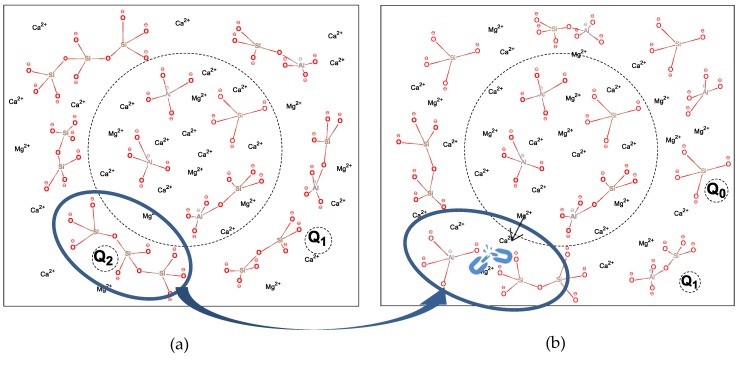
Schematic diagram of the effect of S/A on the structure of amorphous networks: (**a**) slag with higher S/A; (**b**) slag with lower S/A. Note: This figure just serves as a schematic diagram with rarely a consideration of the real structure of a tetrahedron, the actual distribution state of a metal cation, and the charge balance.

**Figure 5 materials-13-01462-f005:**
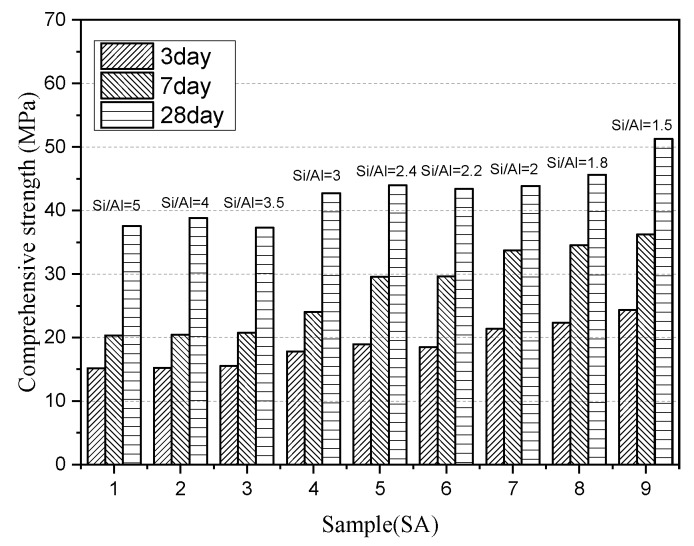
Compressive strength of slag with different S/A at 3 d, 7 d and 28 d ages.

**Figure 6 materials-13-01462-f006:**
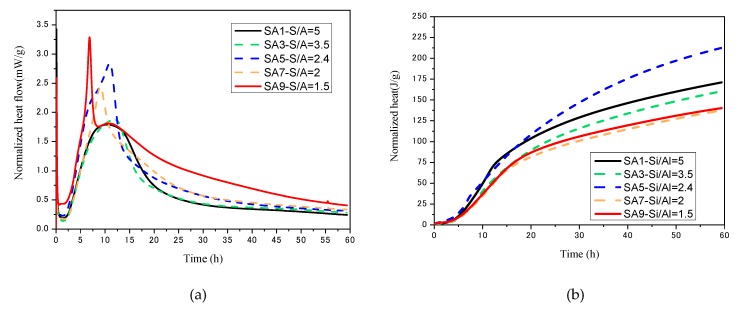
The heat release of slag-cement paste changes with age: (**a**) normalized heat flow; (**b**) accumulated heat release.

**Figure 7 materials-13-01462-f007:**
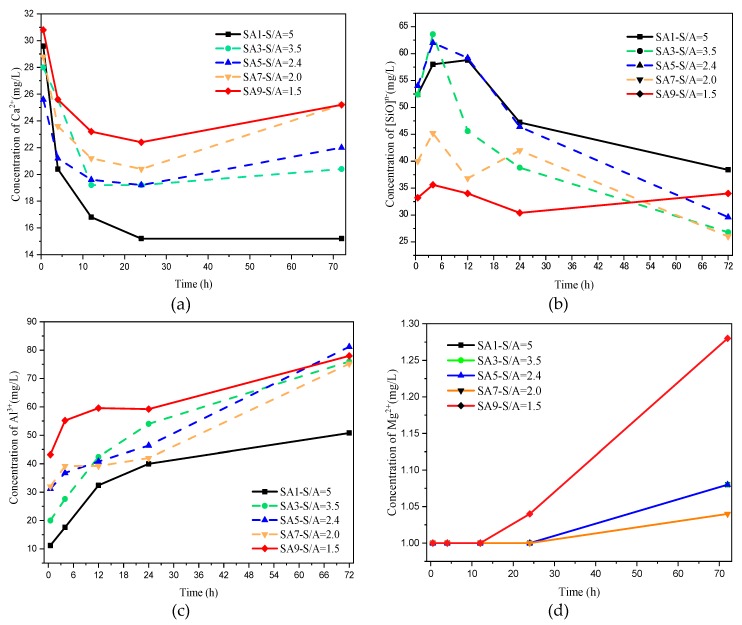
Early ion dissolution characteristics of slag with different S/A in alkaline environment: (**a**) ion dissolution characteristic of Ca^2+^; (**b**) ion dissolution characteristic of [SiO]^n−^; (**c**) ion dissolution characteristic of Al^3+^; (**d**) ion dissolution characteristic of Mg^2+^.

**Figure 8 materials-13-01462-f008:**
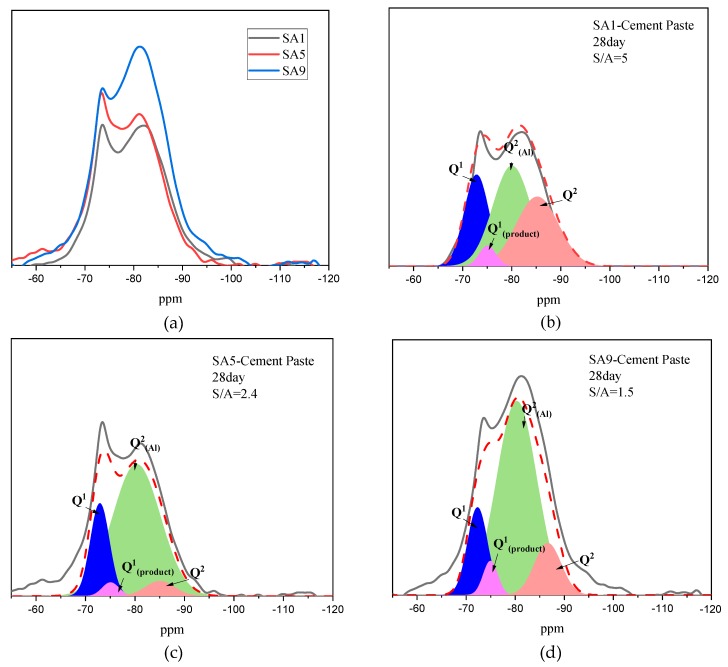
The evolution of the polymerization state of the silicon tetrahedron at a 3-day age: (**a**) stacking chart; (**b**) SA1; (**c**) SA5; (**d**) SA9.

**Figure 9 materials-13-01462-f009:**
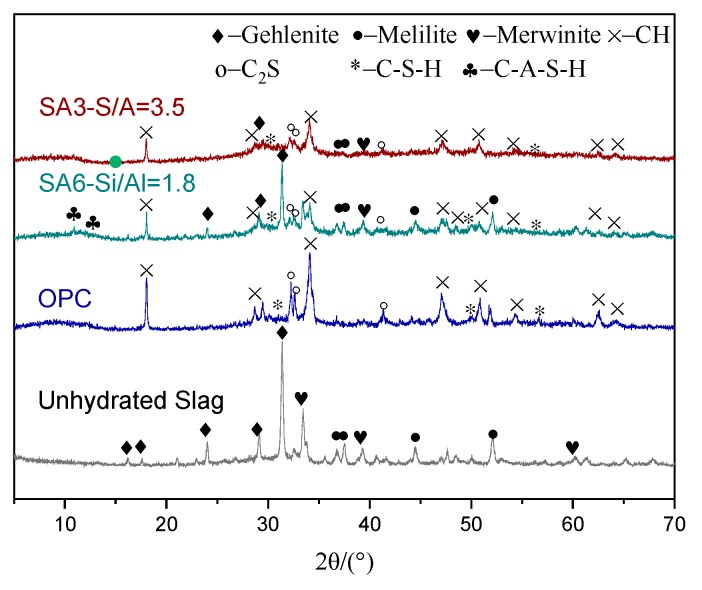
The evolution of the polymerization state of silicon tetrahedron at a 3-day age: (**a**) stacking chart; (**b**) SA1; (**c**) SA5; (**d**) SA9.

**Figure 10 materials-13-01462-f010:**
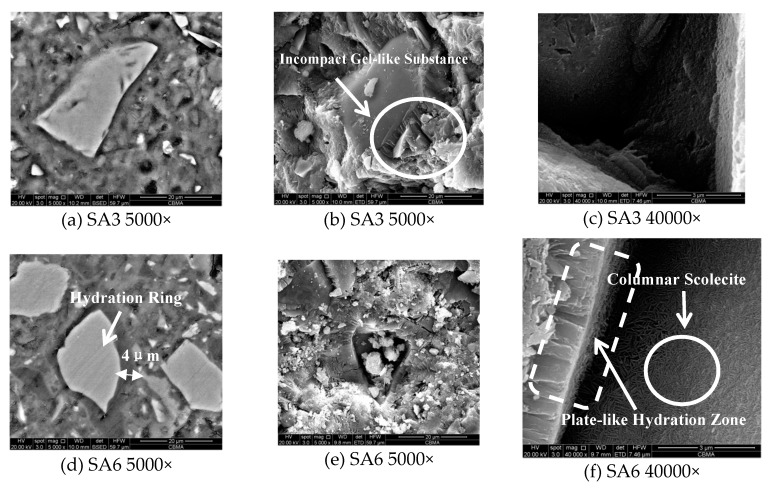
Micro-morphology of slag with different S/A in the paste at 28 days of hydration: (**a**) SA3 5000×; (**b**) SA3 5000×; (**c**) SA3 40000×; (**d**) SA6 5000×; (**e**) SA6 5000×; (**f**) SA6 40000×.

**Figure 11 materials-13-01462-f011:**
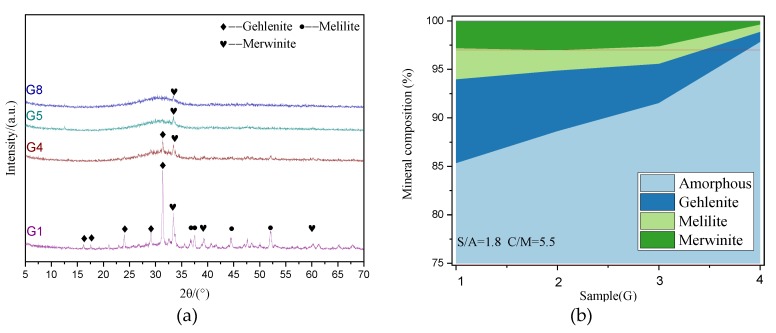
Mineral composition of slag with different amorphous contents: (**a**) XRD patterns; (**b**) quantitative analysis.

**Figure 12 materials-13-01462-f012:**
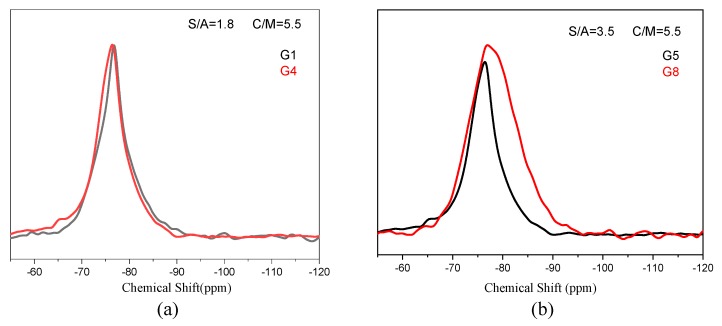
^29^Si MAS NMR spectra of slag with different amorphous content: (**a**) unhydrated G1 and G4; (**b**) unhydrated G5 and G8.

**Figure 13 materials-13-01462-f013:**
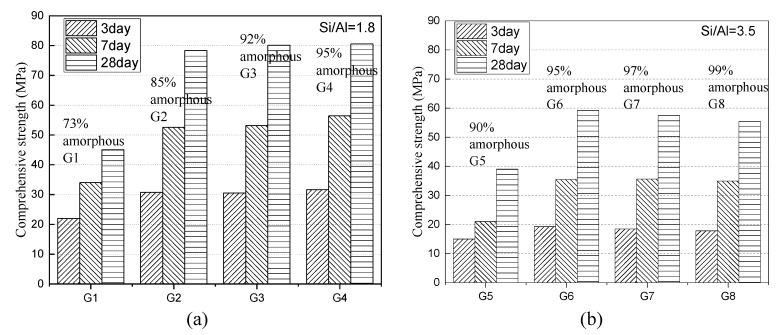
Compressive strength of blending paste with different amorphous contents slag at 3 d, 7 d, and 28 d ages.

**Table 1 materials-13-01462-t001:** Chemical composition of slag with different proportions of network former.

No.	CaO (wt.%)	SiO_2_ (wt.%)	Al_2_O_3_ (wt.%)	MgO (wt.%)	S/A	C/M	(C + M)/(S + A)
SA1	39.96	36.98	9.24	7.43	5.0	5.5	1.03
SA2	39.96	35.95	10.27	7.43	4.0	5.5	1.03
SA3(G5)	39.96	34.67	11.56	7.43	3.5	5.5	1.03
SA4	39.96	32.60	13.62	7.43	3.0	5.5	1.03
SA5	39.96	27.73	18.49	7.43	2.4	5.5	1.03
SA6	39.96	32.60	13.62	7.43	2.2	5.5	1.03
SA7	39.96	31.78	14.44	7.43	2.0	5.5	1.03
SA8(G1)	39.96	30.81	15.41	7.43	1.8	5.5	1.03
SA9	39.96	29.71	16.51	7.43	1.5	5.5	1.03

Note: In the table, S, A, C, and M represent SiO_2_, Al_2_O_3_, CaO, and MgO, respectively, and the proportions in the first row are all mass ratios remain in the same format in this paper.

**Table 2 materials-13-01462-t002:** Chemical composition of slag mixed with trace BaO.

No.	BaO (wt.%)	No.	BaO (wt.%)
SA8(G1)	0	SA3(G5)	0
G2	0.25	G6	0.25
G3	0.5	G7	0.5
G4	0.75	G8	0.75

**Table 3 materials-13-01462-t003:** MAS NMR operating parameters.

**Parameter**	**Pulse Program** **(PULPROG)**	**Time Domain Data Size** **(TD)**	**Number of Scans** **(NS)**	**Receiver Gain** **(RG)**
value	hpdec	409,600	912	1030
**Parameter**	**Acquisition Time** **(AQ)**	**Sweep Width in Hz** **(SWH)**	**Dwell Time** **(DW)**	**Relaxing Delay** **(D1)**
value	0.0425 (s)	48067 (Hz)	10.4 (μs)	60 (s)

**Table 4 materials-13-01462-t004:** Quantitative analysis of Q^n^ in the paste of variable slag.

No.	Q^1^	Q^1^_(__product)_	Q^2^_(Al)_	Q^2^
SA1	24%	4%	41%	32%
SA5	20%	3%	72%	5%
SA9	15%	4%	68%	12%

**Table 5 materials-13-01462-t005:** Quantitative analysis of Q^n^ in the unhydrated slag with different amorphous content.

No.	Q^0^	Q^1^	Q^2^
G1	42%	37%	20%
G4	41%	36%	22%
G5	15%	50%	35%
G8	0%	50%	50%

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
