# Peer review of "Effect of the Formation of Amorphous Networks on the Structure and Hydration Characteristics of Granulated Blast Furnace Slag"

_materials, 2020, doi:10.3390/ma13061462_

Round 1
Reviewer 1 Report
The paper studies the effect of the formation of amorphous network on the structure and hydration characteristics of granulated blast furnace slag. Technical soundness of the paper is good. Relevance to the engineering community, with respect to present works is important. Originality of the tests conducted is good. This paper presents a specific, easily identifiable advance in knowledge. It is applicable and useful to the profession. The title and abstract accurately describe the contents. All references are pertinent and complete. In abstract part of the article stated main aim clearly. Classifications in tables and figures clearly represent experimental studies conducted before. Each figure and table is necessary to the understanding of the conclusions. The results are soundly interpreted and related to existing knowledge on the topic. The conclusions are sound and justified. They follow logically from data presented. All elements of the manuscript relate logically to the study's statement of purpose. I suggest some questions before its final acceptance:
- Authors must indicate in the abastract what is S/A
- The use of capital letters and punctuation must be corrected
- L.39 When authors say that chemical composition is similar or not and this problem is to some extent limits the quality level of product, the may explain why and include some references about this question.
- Please try to eliminate those multiple references with too many citations each.
Author Response
Thanks very much for your kind work and helpful comments and we have modified the paper according to your suggestions.
Point 1: Authors must indicate in the abstract what is S/A
Response 1: We are very sorry for our negligence of using an ambiguous abbreviation in the abstract, and we have fixed this problem.
Point 2: The use of capital letters and punctuation must be corrected
Response 2: We have enhanced the uniformity of the use of capital letters of S/A, [SiO4]4- tetrahedron, [AlO4]5- tetrahedron and Q2(Al).
Point 3: L.39 When authors say that chemical composition is similar or not and this problem is to some extent limits the quality level of product, the may explain why and include some references about this question.
Response 3:Thanks to the reviewer for pointing out this problem, which is a missing of our work. As the reviewers suggested, we have added the relationship of the quality of slag and its utilization in the revised version, and a more detailed explanation of the chemical effect on the slag has been added to the new section of the discussion part.
Point 4: Please try to eliminate those multiple references with too many citations each.
Response 4:We have eliminated the redundant according to the reviewer’s suggestion

Reviewer 2 Report
The paper entitled " Effect of the formation of amorphous network on the structure and hydration characteristics of granulated blast furnace slag" has great potential to be interesting for scientific society. The topic is original and present is important in the use in practice.
Generally, the text of the paper is well written. The presentation is clear and technically correct. There are quite interesting research results. The methods of research are solid and the materials are well documented.
Introduction is interesting, contains quite a lot of references to literature, and finally contains the specific purpose of the authors' research.
However, the article structure presents some shortcomings that must be addressed before publication.
I have comments:
- In point 2.1.3 and Figure 1 is description specific surface area of 450m2/Kg. The unit is m2/kg (small letter k).
- What are NMR operating parameters? Please explain the designations in the table 3.
- Figure 5 please correct - with or without a frame.
- There isn’t discussion in this paper. What the authors showed in point 4 are the conclusions of the research work. Discussion needs to be written in a separate section 4, and in section 5 - conclusions. It really needs to be revised and much improved, needs analysis of results in relation to literature. In its current form, the paper looks like a boring technical report (despite important and interesting research), there are no analysis in relation to existing research or scientific articles.
This part of the article is important to present:
- the background and context
- related studies and actual knowledge !!
- why this study is pertinent and how could potentially improve the knowledge.
According my suggestion the paper needs major restructuration and complete discussion.
I recommend the paper to publish after major revisions.
Author Response
Thanks very much for your kind work and helpful comments and we have modified the paper according to your suggestions.
Point 1: In point 2.1.3 and Figure 1 is description specific surface area of 450m2/Kg. The unit is m2/kg (small letter k).
Response 1: We are very sorry for our incorrect writing of unit notation in the paper, and we have fixed this problem.
Point 2: What are NMR operating parameters? Please explain the designations in the table 3.
Response 2: As Reviewer suggested that we have complemented the full name of NMR operating parameters, and deleted some unnecessary ones. The remaining parameters are matched with the MAS detect of slag and are obtained by a repeated adjustment in the process of the experiment to get a spectrum of high signal strength and low signal to noise ratio. The removed parameters are the default parameters of the test and are not changed drastically during the test. Some unit is attached to the parameters and some are not because they are originally lack of unit. Very thanks for the reviewer to point out this problem to help us to avoid the ambiguity if the experiment will need to repeat.
Point 3: - Figure 5 please correct - with or without a frame.
Response 3:We have made corrections according to the Reviewer’s comments.
Point 4:
There isn’t discussion in this paper. What the authors showed in point 4 are the conclusions of the research work. Discussion needs to be written in a separate section 4, and in section 5 - conclusions. It really needs to be revised and much improved, needs analysis of results in relation to literature. In its current form, the paper looks like a boring technical report (despite important and interesting research), there are no analysis in relation to existing research or scientific articles.
Response 4:This is a very important piece of comments which reminds me of what my academic writing teacher said that the difference between the report and the paper is whether the results are discussed.I feel very sorry for my negligence and we have re-written this part according to the reviewer’s suggestion. In the rearranged part 4, we have summarized some of the latest results of related research and the relationship between our work and them, and point out which part is the extend of previous. The significance of our research results to practical engineering is also illustrated in this part. In the end, the previous part 4 has changed to part 5 that the conclusion of this paper.

Reviewer 3 Report
The authors present a paper on the effect of the formation of amorphous network on the structure and hydration characteristics of granulated blast furnace slag.
The theme is interesting; however it is not new, there are already some published papers regarding the properties and use of granulated blast furnace slag. From this point, it is essential that the authors mention the level of innovation of the proposed work, that is, they must clearly present the contribution of this work to the current state of knowledge on the topic.
In addition, the work plan is well organized and clearly presented. The presentation and analysis of results is also well structured with significant results, however some bench-marketing is missing, which is essential for this type of work.
The conclusions are presented clearly and concisely.
In my opinion, the present work has conditions to be considered for publication.
Author Response
Point 1: The theme is interesting; however it is not new, there are already some published papers regarding the properties and use of granulated blast furnace slag. From this point, it is essential that the authors mention the level of innovation of the proposed work, that is, they must clearly present the contribution of this work to the current state of knowledge on the topic.
Response 1: Thank you for the reviewers’ comments which are all valuable and very helpful for revising and improving our paper. As suggested, we have added the innovation of our work on the abstract, and discuss the connection and extension to the previous research of our work in a new part of the discussion. Although the structure detection technologies and means for slag is quite complete, such technologies are not systematically used to detect the slags that have regularly varying chemical composition. The hydration activity of slag is not only related to the chemical composition or to the structure, but it should be analyzed as a correlative system, and our work is trying to study the slag based on such point of view.

Round 2
Reviewer 2 Report
The authors corrected all comments indicated in the review. I accept the paper for publication.
Author Response
Thanks very much for your kind work. On behalf of my co-authors, we would like to express our great appreciation to editor and reviewers.
